# Clinical and biomarker analyses of sintilimab versus chemotherapy as second-line therapy for advanced or metastatic esophageal squamous cell carcinoma: a randomized, open-label phase 2 study (ORIENT-2)

Jianming Xu [1,29 ✉], Yi Li[1,29], Qingxia Fan[2], Yongqian Shu[3], Lei Yang[4], Tongjian Cui[5], Kangsheng Gu[6], Min Tao[7], Xiuwen Wang[8], Chengxu Cui[9], Nong Xu[10], Juxiang Xiao[11], Quanli Gao[12], Yunpeng Liu[13], Tao Zhang[14], Yuxian Bai[15], Wei Li[16], Yiping Zhang[17], Guanghai Dai[18], Dong Ma[19], Jingdong Zhang[20], Chunmei Bai[21], Yunchao Huang[22], Wangjun Liao [23], Lin Wu[24], Xi Chen[25], Yan Yang[26], Junye Wang[27], Shoujian Ji[1], Hui Zhou[28], Yan Wang[28], Zhuo Ma[28], Yanqi Wang[28], Bo Peng[28], Jiya Sun[28] & Christoph Mancao[28]

This randomized, open-label, multi-center phase 2 study (NCT03116152) assessed sintilimab, a PD-1 inhibitor, versus chemotherapy in patients with esophageal squamous cell carcinoma after first-line chemotherapy. The primary endpoint was overall survival (OS), while exploratory endpoint was the association of biomarkers with efficacy. The median OS in the sintilimab group was significantly improved compared with the chemotherapy group (median OS 7.2 vs.6.2 months; $P = 0.032$; HR = 0.70; 95% CI, 0.50–0.97). Incidence of treatment-related adverse events of grade 3–5 was lower with sintilimab than with chemotherapy (20.2 vs. 39.1%). Patients with high T-cell receptor (TCR) clonality and low molecular tumor burden index (mTBI) showed the longest median OS (15.0 months). Patients with NLR < 3 at 6 weeks post-treatment had a significantly prolonged median OS (16.6 months) compared with NLR ≥ 3. The results demonstrate a significant improvement in OS of sintilimab compared to chemotherapy as second-line treatment for advanced or metastatic ESCC.

A full list of author affiliations appears at the end of the paper.

Esophageal squamous-cell carcinoma (ESSC) is the pre-dominant subtype of esophageal cancer in Asian populations, accounting for 90% of cases[1,2]. More than half of global ESSC cases occur in China[3]. The patients with ESCC have a poor prognosis with a 5-year overall survival rate ≤15%[4]. As the second-line therapy, single-agent chemotherapy is an established option with limited clinical benefit and common toxicities[5–11]. Since 2019, several PD-1 inhibitors are approved as second-line therapy for ESCC[12–14]. Currently the most prominent biomarker for PD-1/PD-L1 treatment options is PD-L1 expression, but this biomarker remains to be imperfect in identifying benefitting ESCC patients. It is therefore of greatest importance to identify alternative biomarkers.

Sintilimab is a humanized, monoclonal antibody against PD-1 that has been approved by NMPA for the monotherapy of relapsed or refractory classical Hodgkin lymphoma, the first-line treatment of nonsquamous and squamous NSCLC in combination with chemotherapy, and the first-line treatment of hepatocellular carcinoma in combination with bevacizumab[15–18].

Here, we conducted the ORIENT-2 study to compare the efficacy and safety of sintilimab vs. chemotherapy as second-line treatment for advanced or metastatic ESCC. In addition, it was expected to observe the alternative predictive biomarkers for these patients treated with sintilimab through neutrophil-to-lymphocyte ratio (NLR), RNA sequencing (RNA-seq) and circulating-tumor DNA (ctDNA).

## Results

**Patient characteristics.** Between May 16, 2017 and August 30, 2018, 253 patients were screened and 190 eligible patients were randomly assigned to receive either sintilimab or chemotherapy ($n = 95$ per group, Fig. 1). In total, 94 patients in the sintilimab

group and 87 patients (15 with paclitaxel and 72 with irinotecan) in the chemo group received at least one dose of the assigned treatment (Fig. 1). Both the demographics and disease characteristics at baseline were generally balanced between the two groups (Table 1).

**Efficacy.** As of the cutoff date (August 2, 2019), the median duration of follow-up was 7.2 months (range, 3.5–12.4) for the sintilimab group and 6.2 months (range, 3.3–10.2) for the chemo group. Overall, 69 (72.6%) patients in the sintilimab group and 81 (85.3%) patients in the chemo group had death. The median overall survival (OS) was 7.2 months (95% confidence interval [CI], 5.8–9.7) in the sintilimab group compared with 6.2 months (95% CI, 5.4–7.9) in the chemo group ($P = 0.032$; stratified hazard ratio [HR], 0.70; 95% CI, 0.50–0.97). Meanwhile, a delayed separation of the survival curves was observed (Fig. 2a). The $P$-values of the weighted log-rank test were less than 0.01 from the Fleming–Harrington (0, 0.2), (0, 0.5), and (0, 1), indicating the delayed effect of the significant OS benefit with sintilimab over chemo. It was thus assumed that sintilimab did not increase the risk of death if used at an earlier time according to the Fleming–Harrington (1, 0) ($P = 0.242$) (Table S3). The estimated 12-month OS rate of patients with sintilimab vs. chemo was demonstrated to be 37.4% vs. 21.4% (Fig. 2a). Moreover, the restricted mean survival time (RMST) at 9, 12, 15, and 18 months after randomization in the sintilimab and chemo groups was 6.3 vs. 6.0 months, 7.5 vs. 6.9 months, and 8.6 vs. 7.4 months, 9.2 vs. 7.6 months.

In total, 78 (82.1%) patients in the sintilimab group and 73 (76.8%) in the chemo group had progressive disease or death. However, the difference in median progression-free survival (PFS) per RECIST v1.1 was not significant ($P = 0.979$; stratified

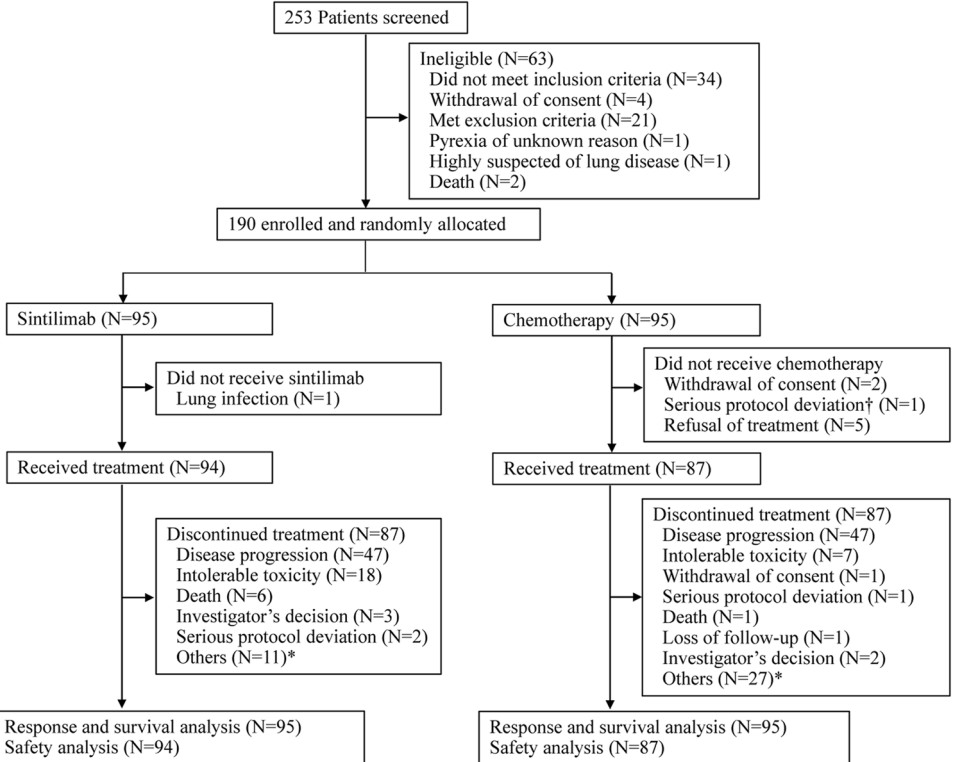

**Fig. 1 CONSORT diagram of patient flow.** This figure shows reasons for exclusion from the study and the numbers of patients included in the analyses. †, unqualified pathological type; *, patients refused to receive treatment, but received follow-up. Treatment discontinuation occurred due to prespecified conditions such as serious protocol deviation, using prohibited drug in the study, loss of follow-up, life-threatening adverse events and other unacceptable toxicities.

**Table 1 Baseline patient characteristics.**

| Characteristic | Sintilimab (N = 95) | Chemo (N = 95) |
|---|---|---|
| *Age, years* | | |
| Median (IQR) | 60 (54-64) | 60 (54-64) |
| <65 | 74 (77.9%) | 75 (78.9%) |
| ≥65 | 21 (22.1%) | 20 (21.1%) |
| *Sex* | | |
| Male | 88 (92.6%) | 84 (88.4%) |
| Female | 7 (7.4%) | 11 (11.6%) |
| *ECOG PS* | | |
| 0 | 23 (24.2%) | 23 (24.2%) |
| 1 | 72 (75.8%) | 72 (75.8%) |
| *Pathological diagnosis* | | |
| Squamous cell carcinoma | 95 (100%) | 95 (100%) |
| *Disease stage[a]* | | |
| IIIA | 2 (2.1%) | 3 (3.2%) |
| IIIB | 1 (1.1%) | 0 |
| IIIC | 4 (4.2%) | 3 (3.2%) |
| IV | 86 (90.5%) | 89 (93.7%) |
| *Site of metastases* | | |
| Lymph node | 67 (70.5%) | 64 (67.4%) |
| Liver | 27 (28.4%) | 33 (34.7%) |
| Lung | 37 (38.9%) | 46 (48.4%) |
| Bone | 6 (6.3%) | 6 (6.3%) |
| Other | 31 (32.6%) | 32 (33.7%) |
| *PD-L1 expression* | | |
| Combined Positive Score <1 | 11 (11.6%) | 17 (17.9%) |
| Combined Positive Score ≥1 | 63 (66.3%) | 50 (52.6%) |
| Combined Positive Score <10 | 34 (35.8%) | 47 (49.5%) |
| Combined Positive Score ≥10 | 40 (42.1%) | 20 (21.1%) |
| Tumor Proportion Score <1% | 44 (46.3%) | 43 (45.3%) |
| Tumor Proportion Score ≥1% | 30 (31.6%) | 24 (25.3%) |
| Tumor Proportion Score <10% | 60 (63.2%) | 53 (55.8%) |
| Tumor Proportion Score ≥10% | 14 (14.7%) | 14 (14.7%) |
| Not evaluable | 21 (32.6%) | 28 (33.7%) |
| *Previous treatment* | | |
| First-line chemotherapy | 84 (88.4%) | 85 (89.5%) |
| Prior to adjuvant chemotherpy (Disease progression within 6 months) | 11 (11.6%) | 10 (10.5%) |
| Radiotherapy | 50 (52.6%) | 58 (61.1%) |
| Surgery | 58 (61.1%) | 47 (49.5%) |

Data are n (%), unless otherwise specified.
ECOG PS Eastern Cooperative Oncology Group performance status.
[a]The disease stage is the stage of clinical TNM at the screening time accroding to AJCC 7th.

HR = 1.00; 95% CI, 0.72–1.39). The median PFS was 1.6 months (95% CI, 1.5–2.8) in the sintilimab group and 2.9 months (95% CI, 2.6–3.6) in the chemo group (Fig. 2b). The 12-month PFS rate for patients treated with sintilimab vs. chemo was 10.4% vs.1.7% (Fig. 2b), with the median PFS in the sintilimab group per iRECIST being 3.8 months (95% CI, 2.9–6.5).

After initial disease progression per RECIST v1.1, 37 patients in the sintilimab group continued to receive sintilimab (continuous subgroup), whereas treatment was discontinued for 28 patients (noncontinuous subgroup). A post hoc analysis of OS in the above subgroups indicated that the median OS of patients with continuous treatment was dramatically improved over those with discontinued treatment (median OS, 12.6 [95% CI, 7.2–16.7] vs. 6.2 months [95% CI, 3.5–8.9]; P = 0.008; HR = 0.45; 95% CI, 0.24–0.82; Fig. 2c).

Based on RECIST v1.1, a double overall response rate (ORR) in the sintilimab group (ORR, 12.6%; 95% CI, 6.7–21.0) was achieved in comparison with the chemo group (ORR, 6.3%; 95% CI, 2.4–13.2; odds ratio [OR], 2.15; 95% CI, 0.77–5.98). The

median duration of response (DOR) of patients in the sintilimab group was longer than the chemo group (8.3 months [95% CI, 2.9–20.9] vs. 6.2 months [95% CI, 4.6–8.4], P = 0.345). However, both groups showed a similar disease-control rate (DCR) (44.2% vs. 43.2%, Table 2).

In the subgroup analysis for OS, the HRs favored sintilimab over chemo for the following five subgroups (sintilimab vs. chemo): age ≤65 years, male, previous treatment with paclitaxel, Eastern Cooperative Oncology Group performance-status (ECOG PS) score of 1, and current smokers (Fig. 3). However, no obvious differences were noted in either OS or PFS between the groups across PD-L1-expression subgroups (P > 0.05, Fig. 3 and Fig. S1).

Health-related quality of life was assessed in 87 patients in the sintilimab group and 84 patients in the chemo group at baseline. Based on the change from baseline, patients in the sintilimab group presented with better health-related quality of life from patients in the chemo group (Table S4).

**Safety**. The median duration of treatment with sintilimab, paclitaxel, and irinotecan was 11.9 (range, 3–112), 11.9 (range, 3–42), and 7.0 (range, 3–30) weeks, respectively. The median relative dose intensity was 100.0% (range, 66.7–107.7), 96.5% (range, 80.7–104.5), and 98.2% (range, 55.6–105.6), respectively.

Treatment-emergent adverse events (TEAEs) of any grade occurred in 88 (93.6%) patients with sintilimab, and 81 (93.1%) patients with chemo. Treatment-related adverse events (TRAEs) were reported in 54.3% of patients in the sintilimab group and 90.8% of patients in the chemo group (Table 3). The most common TRAEs were hypothyroidism (12.8%), pulmonary inflammation (10.6%), anemia (8.5%), and decreased white-blood cell (WBC) counts (8.5%) in the sintilimab group. The incidence of serious TRAEs was similar between the sintilimab (18.1%) and chemo (20.7%) groups. The most common serious TRAEs were pulmonary inflammation (7.4%) and abnormal hepatic function (4.3%) in the sintilimab group. However, the incidence of TRAEs of grade 3 or worse was lower in the sintilimab group (20.2%) than in the chemo group (39.1%). The most common grade 3 or worse TRAEs were pulmonary inflammation (5.3%) and elevated lipase levels (4.3%) in sintilimab group. The number of patients discontinuing treatment because of TRAEs was 13 (13.8%) and 7 (8.0%) in the sintilimab and chemo groups, respectively. In addition, three (3.2%) deaths that occurred in the sintilimab group were attributed to treatment-related adverse events, including upper gastrointestinal bleeding, pneumonitis, and lung infection.

Additionally, 29 (30.9%) patients in the sintilimab group experienced immune-related adverse events (irAEs), with pulmonary inflammation (9.6%) and hypothyroidism (8.5%) occurring most frequently (Table 3). Only 8 (8.5%) patients had irAEs of grade 3 or worse.

**Biomarker analyses**. In the sintilimab group, a post hoc analysis of efficacy in association with NLR was performed. Patients with NLR < 3 were assigned into the low NLR group and those with NLR ≥ 3 were assigned into the high NLR group. In total, 81 patients had evaluable NLR at baseline. Patients with a low NLR at baseline showed a significant improvement on OS (HR 0.54, P = 0.019; median 14.0 vs. 6.2 months) and PFS (HR 0.47, P = 0.002; median 2.9 vs. 1.5 months) compared with those with a high NLR at baseline. In total, 65 patients had evaluable NLR at 6 weeks post treatment. Notably, the survival benefit was also significant in patients with low versus high NLR, with a median OS of 16.6 vs. 6.2 months (HR 0.19, P < 0.001) and a median PFS of 4.3 vs. 2.3 months (HR 0.47, P = 0.006; Fig. 4).

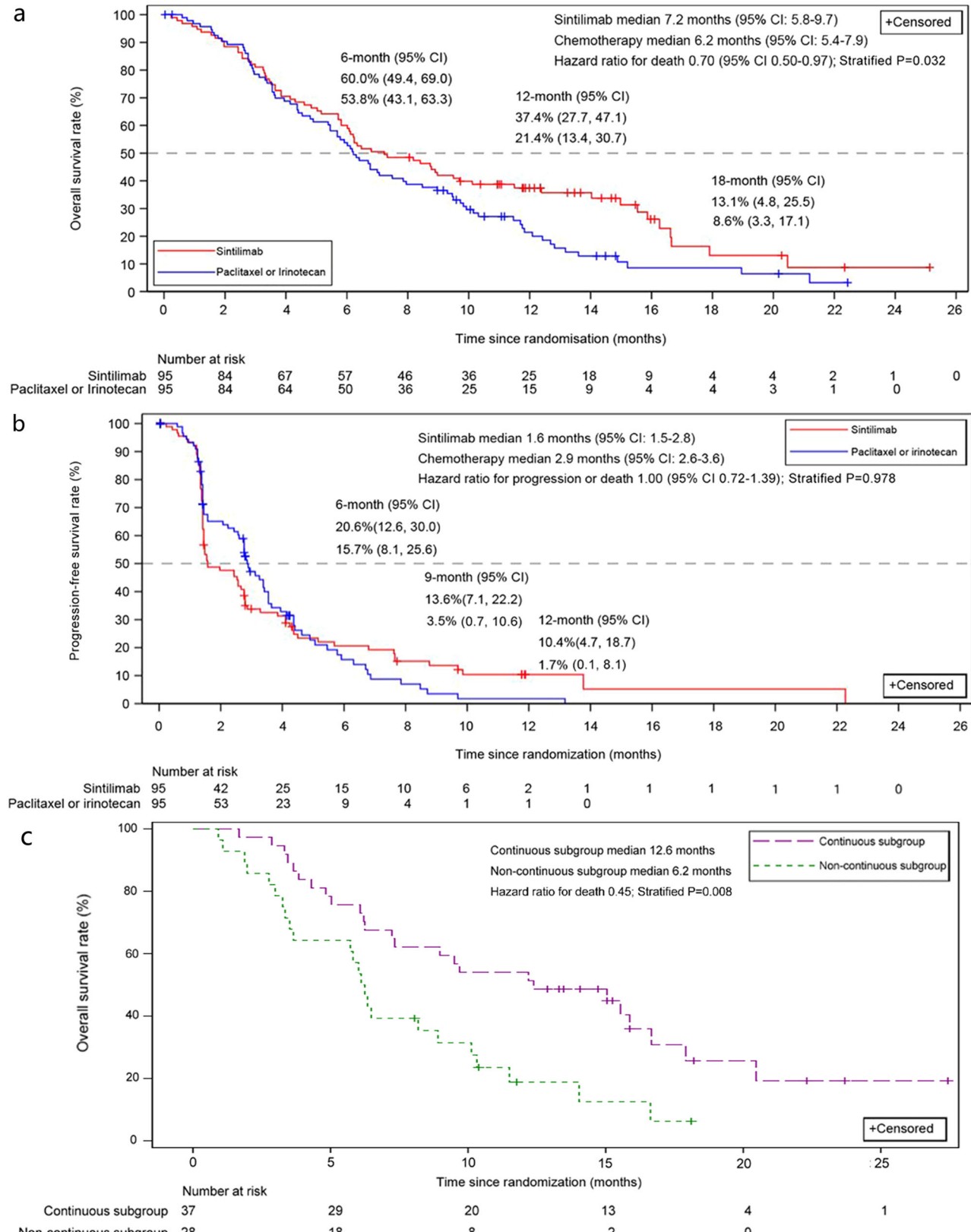

**Fig. 2 Kaplan–Meier plots of survival. a** Overall survival in the ITT population; **b** Progression free survival in the ITT population. **c** Overall survival in the continuous and non-continuous subgroups of the sintilimab group. Log-rank test stratified by Eastern Cooperative Oncology Group performance-status (ECOG PS) score was used (2-sided).

**Table 2 Tumor response to treatment.**

| | Sintilimab (*N* = 95) | Chemo (*N* = 95) | Odds ratio (95% CI) |
|---|---|---|---|
| Best overall response per RECIST v1.1 | | | |
| CR | 0 | 0 | – |
| PR | 12 (12.6%) | 6 (6.3%) | – |
| SD | 30 (31.6%) | 35 (36.8%) | – |
| PD | 41 (43.2%) | 26 (27.4%) | – |
| ORR (CR + PR), % [95% CI] | 12 (12.6%) [6.7%, 21.0%] | 6 (6.3%) [2.4%, 13.2%] | 2.150 (0.770, 5.998) |
| DCR (CR + PR + SD), % [95% CI] | 42 (44.2%) [34.0%, 54.8%] | 41 (43.2%) [33.0%, 53.7%] | 1.045 (0.586, 1.863) |
| Median TTR, month (95% CI) | 1.5 (1.3, 2.8) | 1.4 (1.3, 2.8) | – |
| Median DOR, month (95% CI) | 8.3 (2.9, 20.9) | 6.2 (4.6, 8.4) | – |
| Best overall response per iRECIST | | | |
| iCR | 0 | – | – |
| iPR | 14 (14.7%) | – | – |
| iSD | 30 (31.6%) | – | – |
| iUPD | 22 (23.2%) | – | – |
| iCPD | 15 (15.8%) | – | – |
| iORR (iCR+iPR), % [95% CI] | 14 (14.7%) [8.3%, 23.5%] | – | – |
| iDCR (iCR+iPR+iSD), % [95% CI] | 44 (46.3%) [36.0%, 56.8%] | – | – |
| Median iDOR, month (95% CI) | NA (5.6, NA) | – | – |

*RECIST* response evaluation criteria in solid tumors, *iRECIST* modified RECIST v1.1 for immune-based therapeutics, *CR* complete response, *PR* partial response, *SD* stable disease, *PD* progressive disease, *ORR* objective response rate, *DCR* disease control rate, *DOR* duration of response, *TTR* time to response; *iCR*, *iPR*, *iSD*, *iORR* and *iDCR* demote CR, PR, SD, ORR and DCR per iRECIST, respectively; *iUPD* unconfirmed PD per iRECIST, *iCPD* confirmed PD per iRECIST, *CI* confidence interval, *NA* not available.

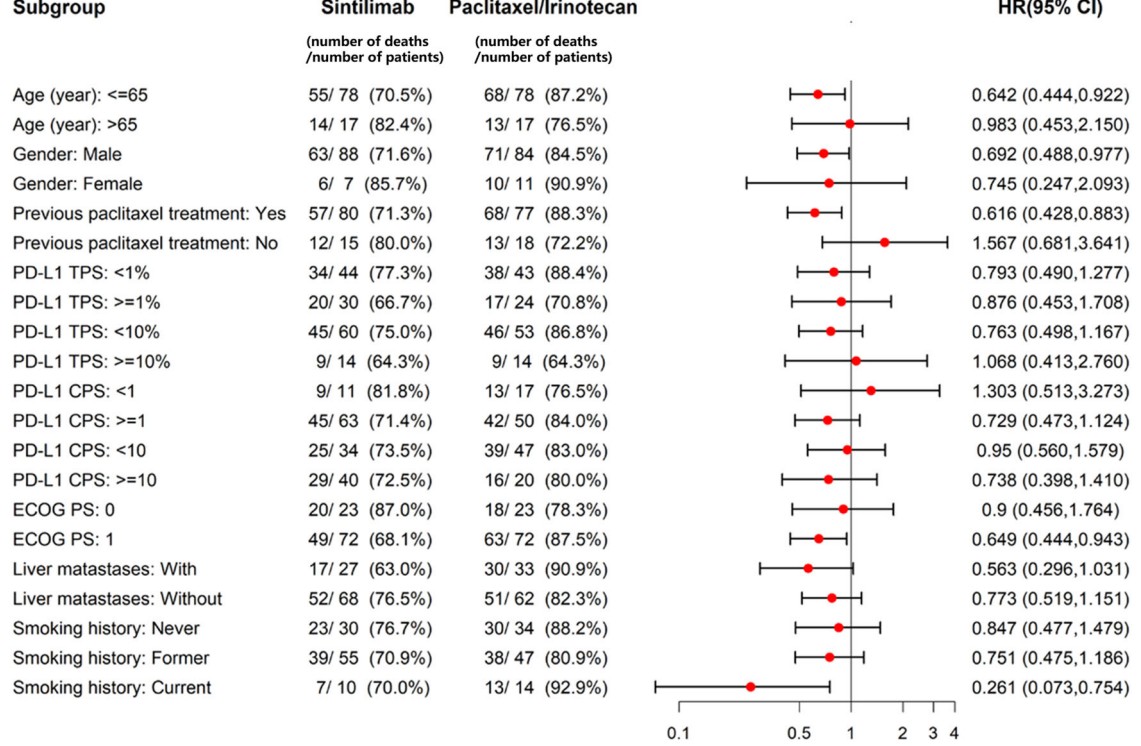

**Fig. 3 Forest plot for subgroup analyses of overall survival.** Dots represent the cohort-specific hazard ratios with error bars corresponding to 95% CI bounds, which were calculated by using the univariate Cox regression model. ECOG PS Eastern Cooperative Oncology Group performance status, HR hazard ratio, CI confidence interval.

In total, 118 patients (*n* = 64 in the sintilimab group and *n* = 54 in the chemo group) had qualifiable RNA-seq data for downstream analysis from tumor tissue. We analyzed an outcome association of 28 immune-cell populations and overall, only few potential positive correlations of immune-cell infiltrations with PFS but not OS could be observed. Only a significant correlation with PFS could be established for high infiltration of T-follicular helper cells (raw *P* = 0.007; HR, 0.46) and activated B cells (raw *P* = 0.030; HR, 0.54) in the sintilimab group (Table S1, Fig. 5).

The two signatures of T-follicular helper cells and activated B-cells were moderately correlated (raw *P* < 0.0001; Spearman's correlation, 0.48).

The outcome association of 45 signaling pathways was analyzed. Based on raw *p*-values, three pathways were significantly associated with improved OS only in the sintilimab group (Wnt, Met, and PI3K/Akt), one pathway was significantly associated with improved OS only in the chemo group (glucose metabolism) and one pathway was significantly associated with

**Table 3 Summary of treatment-related adverse events.**

| | Sintilimab (N = 94) | | | Chemo (N = 87) | | |
|---|---|---|---|---|---|---|
| | Any | Grade 1–2 | Grade ≥3 | Any | Grade 1–2 | Grade ≥3 |
| All events | 88 (93.6) | 36 (38.3) | 52 (55.3) | 81 (93.1) | 40 (46.0) | 41 (47.1) |
| *TRAEs* | 51 (54.3) | 32 (34.0) | 19 (20.2) | 79 (90.8) | 45 (51.7) | 34 (39.1) |
| Hypothyroidism | 12 (12.8) | 12 (12.8) | 0 | 1 (1.1) | 1 (1.1) | 0 |
| Pulmonary inflammation | 10 (10.6) | 5 (5.3) | 5 (5.3) | 1 (1.1) | 1 (1.1) | 0 |
| Anemia | 8 (8.5) | 8 (8.5) | 0 | 33 (37.9) | 28 (32.2) | 5 (5.7) |
| WBC count decreased | 8 (8.5) | 7 (7.4) | 1 (1.1) | 42 (48.3) | 28 (32.2) | 14 (16.1) |
| ALT increased | 7 (7.4) | 7 (7.4) | 0 | 6 (6.9) | 6 (6.9) | 0 |
| AST increased | 7 (7.4) | 7 (7.4) | 0 | 5 (5.7) | 5 (5.7) | 0 |
| Amylase increased | 6 (6.4) | 4 (4.3) | 2 (2.1) | 0 | 0 | 0 |
| Cough | 6 (6.4) | 4 (4.3) | 2 (2.1) | 1 (1.1) | 1 (1.1) | 0 |
| Abnormal liver function | 6 (6.4) | 5 (5.3) | 1 (1.1) | 4 (4.6) | 3 (3.4) | 1 (1.1) |
| Neutrophils count decreased | 5 (5.3) | 3 (3.2) | 2 (2.1) | 30 (34.5) | 14 (16.1) | 16 (18.4) |
| Platelet count decreased | 5 (5.3) | 3 (3.2) | 2 (2.1) | 10 (11.5) | 9 (10.3) | 1 (1.1) |
| Fatigue | 5 (5.3) | 5 (5.3) | 0 | 19 (21.8) | 19 (21.8) | 0 |
| Lipase elevated | 4 (4.3) | 0 | 4 (4.3) | 0 | 0 | 0 |
| Diarrhea | 4 (4.3) | 4 (4.3) | 0 | 29 (33.3) | 24 (27.6) | 5 (5.7) |
| Lymphocyte count decreased | 3 (3.2) | 1 (1.1) | 2 (2.1) | 5 (5.7) | 4 (4.6) | 1 (1.1) |
| Hypochloremia | 3 (3.2) | 1 (1.1) | 2 (2.1) | 0 | 0 | 0 |
| Nausea | 3 (3.2) | 3 (3.2) | 0 | 28 (32.2) | 26 (29.9) | 2 (2.3) |
| Proteinuria | 3 (3.2) | 3 (3.2) | 0 | 5 (5.7) | 4 (4.6) | 1 (1.1) |
| Vomiting | 2 (2.1) | 2 (2.1) | 0 | 18 (20.7) | 14 (16.1) | 4 (4.6) |
| Upper gastrointestinal hemorrhage | 2 (2.1) | 0 | 2 (2.1) | 0 | 0 | 0 |
| Lung infection | 2 (2.1) | 0 | 2 (2.1) | 2 (2.3) | 1 (1.1) | 1 (1.1) |
| Decreased appetite | 1 (1.1) | 1 (1.1) | 0 | 17 (19.5) | 15 (17.2) | 2 (2.3) |
| Abdominal pain | 1 (1.1) | 1 (1.1) | 0 | 6 (6.9) | 6 (6.9) | 0 |
| Hypokalemia | 1 (1.1) | 1 (1.1) | 0 | 5 (5.7) | 4 (4.6) | 1 (1.1) |
| Alopecia | 0 | 0 | 0 | 13 (14.9) | 13 (14.9) | 0 |
| Bone marrow failure | 0 | 0 | 0 | 10 (11.5) | 4 (4.6) | 6 (6.9) |
| Hypaesthesia | 0 | 0 | 0 | 6 (6.9) | 4 (4.6) | 2 (2.3) |
| *Immune-related AE* | 29 (30.9) | 21 (22.3) | 8 (8.5) | 0 | 0 | 0 |
| Pulmonary inflammation | 9 (9.6) | 5 (5.3) | 4 (4.3) | 0 | 0 | 0 |
| Hypothyroidism | 8 (8.5) | 8 (8.5) | 0 | 0 | 0 | 0 |
| Rash | 4 (4.3) | 3 (3.2) | 1 (1.1) | 0 | 0 | 0 |
| Abnormal liver function | 3 (3.2) | 3 (3.2) | 0 | 0 | 0 | 0 |
| Lung infection | 2 (2.1) | 1 (1.1) | 1 (1.1) | 0 | 0 | 0 |
| Psoriasis | 2 (2.1) | 2 (2.1) | 0 | 0 | 0 | 0 |
| Diarrhea | 2 (2.1) | 2 (2.1) | 0 | 0 | 0 | 0 |
| Anemia | 2 (2.1) | 2 (2.1) | 0 | 0 | 0 | 0 |

Data are presented in %. Grade 1–2 TRAE listed with an incidence of ≥5% of patients in either treatment group, and grade 3–5 TRAEs with an incidence of ≥2% in either group. IrAEs were occurred in ≥2% of patients in either treatment group. *TRAE* treatment-related adverse event, *irAE* immune-related adverse event, *WBC* white blood cell, *ALT* alanine aminotransferase, *AST* aspartate aminotransferase.

improved OS in both groups (HiF1 signaling, Table S2). A correlation with PFS could only be detected for patients with induced Nrf2 pathway and high oxidative stress in the sintilimab group.

To further explore indicators of clinical benefit in the periphery blood, T-cell receptor (TCR, $n = 94$) and circulating cell-free DNA (cfDNA, $n = 83$) sequencing was performed at baseline for patients in the sintilimab group in order to derive TCR clonality and molecular tumor-burden index (mTBI). Our analyses showed that patients with low mTBI had a significantly longer PFS ($P = 0.018$; HR, 0.55) compared with those with high mTBI. Similar trend was also observed for OS ($P = 0.200$; HR, 0.71). In contrast, TCR clonality alone could not efficiently predict the clinical outcome (Fig. S2). However, the prediction power for OS benefit could be enhanced when TCR clonality was combined with mTBI. Despite the limited sample size, the patients with high TCR clonality and low mTBI had the longest median OS (15.0 months) and PFS (4.1 months), respectively (Fig. 6).

## Discussion

This study revealed that sintilimab monotherapy resulted in an evidently prolonged survival outcome with a favorable safety profile vs. chemotherapy as second-line treatment in Chinese patients with advanced or metastatic ESCC.

Up to now, four similar studies of PD-1 inhibitors have reported encouraging clinical efficacy in patients with ESCC after first-line therapy, including KEYNOTE-181 of pembrolizumab, ATTRACTION-3 of nivolumab, ESCORT of camrelizumab, and RETIONALE-302 of tislelizumab[12–14,19]. In these studies the median OS of PD-1 inhibitor groups are all greater than 7.0 months and the HRs for OS are all lesser than 0.8[12–14,19]. The significant survival benefit of sintilimab over chemotherapy for patients with ESCC was consistent with the other studies. The Kaplan–Meier survival curves for OS were shown to diverge beyond 5 months in favor of sintilimab. This was similar to the results obtained in the other four studies, which was the Kaplan–Meier survival curves for OS diverged beyond 3–6 months[12–14,19].

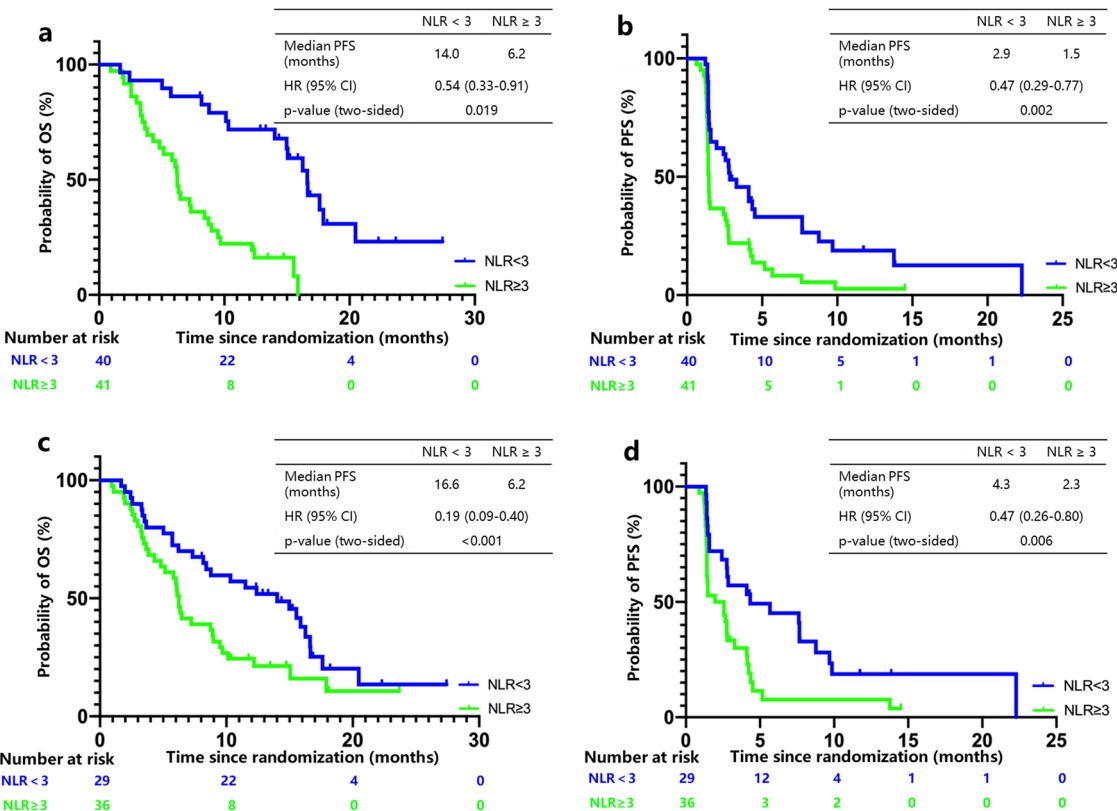

**Fig. 4 Kaplan–Meier plots of survival in high and low neutrophil-to-lymphocyte ratio (NLR) subgroups of the sintilimab group. a** Overall survival with NLR at baseline. **b** Progression-free survival with NLR at baseline. **c** Overall survival with NLR at 6 weeks post treatment. **d** Progression-free survival with NLR at 6 weeks post treatment. HR hazard ratio. *P*-values were based on a two-sided log-rank test.

RMST is a statistical technique recently adopted widely as an alternative to median to provide an intuitive interpretation of the data. It is especially more appealing than median survival time when the data violate proportional hazard assumption when only evaluating median difference could be very misleading. The superior RMST of sintilimab over chemo was increased from 0.3 months at 9 months after randomization to 1.6 months at 18 months after randomization, further suggesting a delayed survival benefit of sintilimab. In addition, *P*-values of different weight coefficients (0, 0.2), (0, 0.5), and (0, 1) from Fleming–Harrington test are less than 0.01, indicating the delayed effect of the significant OS benefit with sintilimab over chemo. Consequently, it is suggested that there are a delayed response and more durable benefit of immunotherapies based on similar studies and different statistical methods. The survival advantage of sintilimab as second treatment for patients with ESCC may be demonstrated via a prolonged follow-up duration.

Moreover, the ORR of patients with sintilimab was twice higher than those with chemo, similar to the findings in the other four studies[12–14,19]. Notably, for patients who were initially evaluated as PD per RECIST v1.1 in the sintilimab group, those continuing to receive sintilimab had a remarkable increase on median OS, compared with those who discontinued treatment. This finding suggested that continuous treatment with sintilimab could have a long-term survival benefit and should be further explored for patients after initial PD by RECIST v1.1.

Sintilimab showed a favorable safety profile over chemo, with a substantially reduced incidence of TRAEs (54.3 vs. 90.8%) and TRAEs of grade 3 or worse (20.2 vs. 39.1%). Most toxic effects were comparable to the historical data on treatment with sintilimab in other indications and to that of treatments with other PD-1 inhibitors in esophagus cancer[12–19]. In particular, three treatment-related deaths, which were due to upper gastro-intestinal bleeding, pneumonitis, and lung infection, occurred in the sintilimab group. Pneumonitis and lung infection were previously reported in PD-1 inhibitor-treated patients with esophagus cancer[12–14,19]. Nevertheless, upper gastrointestinal bleeding is one of the most common fatal causes for ESCC. Upper gastro-intestinal bleeding was considered to be treatment-related, perhaps because this study was an open-label study.

In this study, comprehensive biomarker analyses were performed, including tumor PD-L1 expression, NLR, tumor transcriptome analysis and peripheral blood TCR, and cfDNA analysis.

Even if only the HR for OS in the combined-proportion score (CPS) ≥ 10 subgroups was less than the CPS < 10 subgroups, neither PD-L1-expression level assessed by tumor-proportion score (TPS) nor CPS could predict definitely benefit from the treatment with sintilimab unlike other studies[12–14,19]. That is potentially attributed to limited sample availability.

NLR is an inflammatory-response biomarker for the prognosis of ESCC in patients treated with chemoradiotherapy and immune-checkpoint inhibitors[20–23]. As is known, NLR can reflect differences in the immune status during cancer development and progression[22,23]. A meta-analysis showed that high NLR is associated with worse survival in esophageal cancer, as it is also shown in several solid tumors[24]. Our results revealed that low NLR (<3) at baseline or at 6 weeks post treatment with sintilimab was significantly correlated with a longer OS and PFS. The increased number of neutrophils in the peritumoral inflammatory-cell infiltrate can reduce the antitumor activity of natural-killer (NK) and activated T cells[25,26]. Moreover, many cytokines (e.g., TNF, IL-1, and IL-6) and VEGF produced as a result of neutrophil activation may increase tumor growth[27]. But

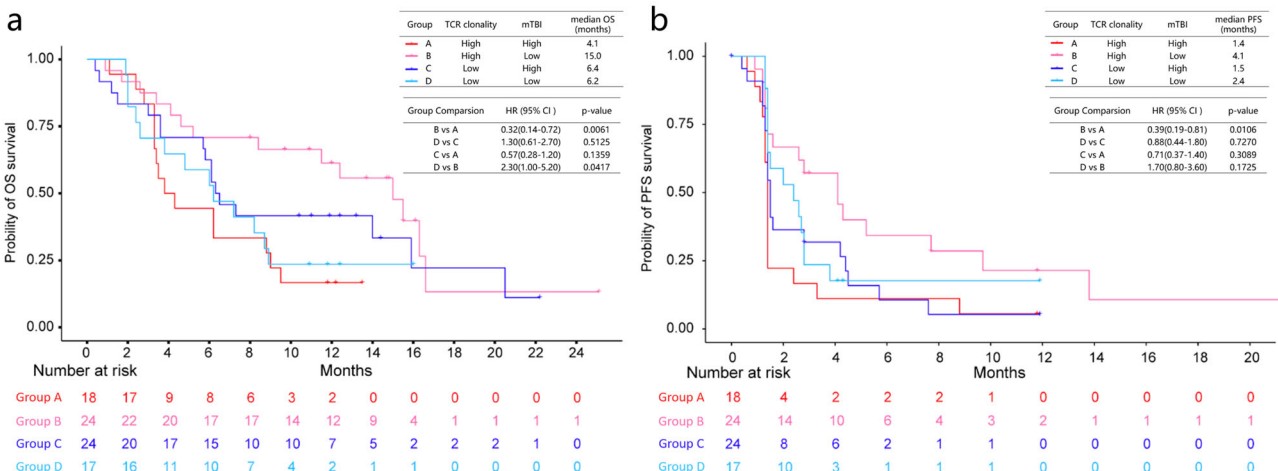

**Fig. 5 Heatmap and forest plots of the association of tumor microenvironment immune-cell signatures and progression-free survival.** Heat map of immune score of 28 immune-cell populations is shown and forest plots (right panel) display the correlation of each immune subtype with progression-free survival. Node position reflects hazard ratio (<1 for favorable outcome with high score in the respective treatment). Only significant correlations are displayed and node size reflects the p-value (the larger the node size, the more significant). P-values were based on a two-sided Wald test. Source data are provided as a Source Data file.

**Fig. 6 Kaplan–Meier plots of survival in different TCR clonality and mTBI subgroups of the sintilimab group. a** Overall survival. **b** Progression-free survival. The high- or low- level groups of TCR clonality or mTBI are split by the respective median value. Group A: high TCR clonality and high mTBI; Group B: high TCR clonality and low mTBI; Group C: low TCR clonality and high mTBI; Group D: low TCR clonality and low mTBI. P-values were based on a two-sided Wald test. Source data are provided as a Source Data file.

the mechanism is still unclear that NLR is related to survival in esophageal cancer and other solid tumors. The test of NLR is very simple and practicable, but the use of NLR in clinical practice is still underestimated.

Although general T-cell tumor infiltration did not show a predictive effect in the analysis of tumor transcriptomes, infiltration of T-follicular helper cells and activated B cells was significantly related to PFS in the sintilimab group and not in the chemo group. Especially, the predictive value of activated B lymphocytes is supported by an emerging role of B cells and tertiary lymphoid structures (TLS) in tumors, which have been recently reported in association with survival, even in tumor with low T-cell infiltration[28–30]. Furthermore, T-follicular helper cells have been shown to play important roles in the generation of TLS in the tumor microenvironment[31], which might be relatively enriched for B-cell signatures and TLS in ESCC[32]. Thus, the potential relevance of TLS in the mode of action of cancer immunotherapy to mount an efficient antitumor response is supported by our study and warrants further investigation. However, due to tumor-specimen limitations, additional analyses to characterize TLS and their spatial distribution within the tumor microenvironment could not be performed. The correlation of several signaling pathways with clinical efficacy is more challenging to interpret. Different reports indicate a potentially context-dependent role for Wnt/β-catenin signaling in CD8 + T-cell differentiation and memory formation[33,34]. Wnt pathway also enhances the proliferation and cytolytic activity of human γδT cells[35]. Autocrine Wnt pathway activation induces vascular normalization and endows endothelial-cell basement membrane the capacity to boost T-cell infiltration[36]. Further evidence shows that, in contrast to early-stage tumors, advanced and metastatic tumors have downregulated Wnt expression. In some cases, endogenous Wnt signaling even generally counteracts tumor progression[37]. Also, the small sample size could contribute to these surprising findings. Such conflicting data clearly underline the need for additional analyses and validation of the data.

In contrast to the analysis of tumor tissue, the investigation of peripheral blood has practical advantages without the limitation of tumor tissue accessibility and size. We found that peripheral TCR clonality at baseline did not predict the efficacy of sintilimab. In addition, we also analyzed mTBI as an indicator of tumor burden at the molecular level[38]. Whereas mTBI is derived from peripheral blood, tumor mutation burden (TMB) is generally derived from mutation assessment in tumor tissue. In contrast to TCR clonality, mTBI index was significantly associated with clinical efficacy in the treatment with sintilimab. However, the predictive value of mTBI in combination with TCR clonality might be further enhanced as patients with low mTBI and high TCR clonality exhibited the longest survival in the sintilimab group. TCR-clonality analyses revealed conflicting results in previous studies, potentially due to the dilution effect of nontumor-specific TCRs since this analysis alone cannot distinguish between tumor vs. nontumor specificity[39–41]. However, a high clonality might be indicative of a preexisting and expanded tumor-specific T-cell response that has been blocked or exhausted. It is intriguing to speculate that such a block is mediated by the PD-1 axis and sintilimab treatment might release this block. However, reinvigoration and subsequent tumor eradication seems to mainly occur in patients with low mTBI. Detailed mechanisms for this correlation need to be examined, but apart from physical barriers in larger tumors it was previously reported that the ratio of T-cell reinvigoration to baseline tumor burden instead of the magnitude of the reinvigoration alone might be a better indicator of response[42,43]. Additionally, it has been reported that skewed ratios of infiltrating suppressive immune cells like myeloid-derived suppressor cells and T regulatory cells vs. T-effector cells, dendritic cells, and natural-killer cells correlate with tumor progression and increased tumor burden. Such cellular changes are accompanied with increasingly immune-suppressive cytokine milieus in the growing tumors[44–48]. Our data further support the notion that an effective mounting of an antitumor response is multifactorial and does not only rely on successful reinvigoration of exhausted T cells, but also on tumor burden and preexisting TCR clonality in addition to previously described predictive biomarkers like PD-L1 expression, TMB, and T-cell infiltration. It remains to be investigated if these factors are independent contributors or surrogates of each other.

This study had several limitations. First, the sample size of patients was relatively small. Second, the open-label design of the study might have influenced the assessment of the incidence of adverse effects. Nevertheless, this design was deemed acceptable because both dose regimens and toxicities in the two groups were disparate. Third, it was hard to evaluate the correlation between the expression of PD-L1 and the survival benefit of sintilimab due to the limited tumor samples available for the assessment of PD-L1 expression.

In conclusion, the results of ORIENT-2 study favored the use of sintilimab over chemo in Chinese patients with advanced ESCC refractory to previous chemotherapies, as it suggested a prolonged survival benefit and a favorable safety profile. Both NLR <3 at 6 weeks post treatment and the combination of high TCR clonality with low mTBI may be potent biomarkers for the prediction of improved OS and PFS in patients with ESCC treated with sintilimab.

## Methods

**Study design and patients**. This was a multicenter, randomized, parallel, open-label phase-2 study conducted at 30 sites in China (Table S5). Patients with histopathologically or cytologically confirmed locally advanced or metastatic ESCC aged between 18 and 75 years old were enrolled, including male and female. Major eligibility criteria were at least one measurable lesion per Response Evaluation Criteria in Solid Tumors (RECIST), version 1.1, Eastern Cooperative Oncology Group (ECOG) performance status (PS) of 0 or 1, eligible to provide a fresh or archived tumor sample, and radiological or clinical evidence of disease progression during or after first-line chemotherapy. Patients were excluded who received prior PD-1 or PD-L1 inhibitors, or received any radiotherapy, immunosuppressive drugs, or the study drug within 4 weeks prior to the first dose. Full inclusion criteria are provided in the Supplementary. The study was performed in accordance with the Declaration of Helsinki and Good Clinical Practice guidelines and was approved by the ethics committee at each site. All patients provided written informed consent prior to enrollment. The trial is registered with ClinicalTrials.gov, number NCT03116152.

**Procedures**. Patients were randomly (1:1) assigned to receive either sintilimab or the investigator's choice of chemo (paclitaxel or irinotecan), using an interactive web-response system with a block size of a mixture of 2, 4, and 6, and with the stratification factor of the ECOG PS score (0 vs. 1). Neither investigators nor patients were blinded to treatment allocation.

After randomization, patients were treated with 200 mg of intravenous sintilimab, once every 3 weeks, or with the investigator's choice of chemo (175 mg/m² intravenous paclitaxel once every 3 weeks, or 180 mg/m² intravenous irinotecan once every 2 weeks), until disease progression, death, unacceptable toxicity, or withdrawal of informed consent.

Tumor assessment was performed by investigators according to RECIST version 1.1 every 6 weeks till 24 weeks after randomization and every 9 weeks thereafter until initiation of new antitumor therapy, disease progression, withdrawal of informed consent, or death. In the sintilimab group, patients with stable status and initial progressive disease per RECIST version 1.1 assessed by the investigator could continue to receive sintilimab. Then tumor assessment was performed according to modified RECIST v1.1 for immune-based therapeutics (iRECIST). Adverse events (AEs) and serious adverse events (SAEs) were monitored throughout the treatment period and for 90 days after the last dose, based on the National Cancer Institute Common Terminology Criteria for Adverse Events (NCI CTCAE v4.03).

**Measurements of PD-L1**. The expression of PD-L1 was assessed via immunohistochemistry (Dako PD-L1 IHC 22C3 pharmDx) using the Autostainer Link 48 (Dako, Carpinteria, CA, USA). PD-L1 protein staining was determined using the tumor-proportion score (TPS) or the CPS.

**Measurements of NLR**. The NLR inflammatory biomarker, which was calculated as neutrophil count/lymphocyte count, was obtained from hematology panels at baseline and 6 weeks post treatment. High and low NLR groups were classified using the cutoff of NLR = 3[49].

**Biomarker analysis using next-generation sequencing and bioinformatics methods**. Quality control of RNAseq raw data was performed using FastQC (version 0.11.8)[50]. Clean reads were obtained through trimming clean reads using Trimmomatic (version 0.36)[51]. STAR aligner (version 2.7.0a) was used to map clean reads to human reference genome (hg38)[52]. HTSeq (version 0.11.4) was used for counting reads mapped to genes[53]. The gene sets defining 28 immune-cell populations were extracted from the previous research[38]. The gene sets of 45 signaling pathways were downloaded from the tumor signaling 360 panel of NanoString (Seattle, Washington, US). The tumor samples of biomarker-evaluable population (BEP) with qualified RNA-seq data were used to conduct downstream analysis. To screen for survival-related genes, based on transcript-per-million (TPM)-normalized gene expression level of each protein-coding gene, the BEP was iteratively split into high or low expression groups ($S_{high\_low}$) by median value of the whole cohort of the respective gene. Then, Cox regression model by the formula Surv (time, event) ~$S_{high\_low}$ in each of the treatment arm was run using the R survival package (version 2.44-1.1), to derive the respective hazard ratio and raw $P$-value[54]. All genes were ranked by raw $P$-value, whereas genes with $P < 0.05$ were considered as potentially survival-related genes. Gene-set signature scores were calculated using the GSVA algorithm (version 1.32.0) based on the gene expression levels of TPM[55]. High and low signature scores were split according to the median value of the whole cohort. Then the same screening procedure as described above was conducted.

Peripheral baseline T-cell receptor (TCR) CDR3 regions and cell-free DNA were sequenced to calculate TCR clonality and molecular tumor-burden index (mTBI) in each sample. The latter is a reflection of the percentage of circulating-tumor DNA (ctDNA) detected in whole cell-free DNA (cfDNA) and is considered to reflect the tumor burden at the molecular level, which is defined as the mean allele fraction of mutations in the trunk cluster as published earlier[38]. Sequencing data of ctDNA were mapped to human reference genome (hg19) by using BWA (version 0.7.17)[56]. GATK (version 4.1.4.1) was used to call variants of each sample[57]. PyClone (version 0.13.1) was used to analyze the clonal-population structures based on baseline-mutation profile[58]. Sequencing reads of TCRβ CDR3 region were merged to obtain contigs using Pear (version 0.9.11)[59], and then contig sequences were aligned to reference TRB V/(D)/J gene sequences (http://www.imgt.org) using MiXCR (version 3.0.4) to determine the TRB V/(D)/J gene segment in each contig[60]. High and low TCR clonality or mTBI of baseline samples were split according to the median value of the whole cohort. Then, Cox regression model by the formula Surv(time, event) ~ $S_{high\_low}$ in each treatment arm was run using the R survival package (version 2.44-1.1)[54], to derive the respective hazard ratio (HR) and raw $P$-value (P). The 'time' represents PFS or OS, 'event' for progression or death event, and '$S_{high\_low}$' for split groups by median value. Survival curves were plotted using the R survminer package (version 0.4.6)[61].

**Outcomes**. The primary endpoint was OS, defined as the time from randomization until death. Surviving patients were censored using the data at the last follow-up. Secondary endpoints were PFS, which was the time from randomization to initial disease progression evaluated by the investigator per RECIST v1.1 or death; ORR, defined as the percentage of patients achieving complete response (CR) or partial response (PR); DCR, which was the percentage of patients with the best overall response of CR, PR, or stable disease (SD); time to response (TTR); and duration of response (DOR). As safety endpoint was considered the incidence of all AEs across the study. Exploratory endpoints were the association between efficacy (OS and PFS) and biomarkers, including PD-L1 expression, mRNA expression, and ctDNA results. The association between efficacy (OS, PFS) and NLR was post hoc analysis. Health-related quality of life (HRQoL) was assessed by the European Organisation for Research and Treatment of Cancer Quality-of-Life Questionnaire-Core 30 (EORTC QLQ-C30), and the five-level version of European Quality of Life-5 Dimensions (EQ-5D-5L) visual analogue scale (VAS), at the first dose, as well as at every radiological evaluation time and at the time of the first safety follow-up.

**Statistical analysis**. The required death-event number was 142, with an expected HR for OS of 0.7 ($\alpha = 0.2$, two-sided) under a statistical power of 80%. Assuming a 21% dropout rate, 180 patients were required.

All efficacy endpoints were assessed in the intention-to-treat (ITT) population that included all randomly assigned patients. The safety profile was assessed in the safety set (SS), which included patients receiving at least one dose of the study drug.

The Kaplan–Meier method was utilized to evaluate the time-to-event endpoints (median OS, PFS, and DOR), and the differences in the survival curves between the two groups were analyzed using the stratified log-rank test by ECOG PS score. A stratified Cox proportional hazard regression model was used to estimate the HR and corresponding 95% confidence interval (CI) of OS and PFS between treatment groups, using the ECOG PS score (0 vs. 1) as a stratification factor. The RMST was evaluated to compare the differences in OS between groups at 9, 12, 15, and 18 months post treatment. The binomial distribution method was used to evaluate the 95% CI of ORR and DCR, and differences were compared using the Cochran Mantel–Haenszel test. Patient-reported health-related quality-of-life outcomes

were analyzed for patients who had a baseline and at least one post-baseline assessment of EORTC QLQ-C30 (general health) and EQ-5D-5L (VAS and index).

To evaluate any factors correlated with the crossing of the survival curves in OS, different treatments were assessed using a weighted log-rank test from the FH $G^{(\rho-\gamma)}$, which accounts for nonproportional hazards ($\rho$, $\gamma = 0,1$; 0, 0.5; 0, 0.2; 1, 0). Considering the potential pseudoprogression for PD-1 inhibitors, a post hoc analysis was performed to compare the difference in OS between patients with and without continuous treatment of sintilimab after initial PD per RECIST v1.1 in the sintilimab group.

Statistical analyses for the clinical part were performed using the SAS software (version 9.4) (Cary, North Carolina, US). The significance level for all endpoints was $\alpha = 0.05$ (two-sided) and the significance threshold was $P < 0.05$. For biomarker identification using bioinformatics, survival analysis was performed using the survival package[43] in R, by which both the $P$-value and hazard ratio between two groups were calculated using the coxph function. Survival curves were plotted using the survminer package[44].

**Reporting summary**. Further information on research design is available in the Nature Research Reporting Summary linked to this article.

## Data availability
All study data are presented in the paper and supplementary files. Source data are provided with this paper, including the baseline patient information, the results of efficacy in all patients, the NLR, and the results of RNA sequencing, TCR clonality, and molecular tumor-burden index (mTBI) in the sintilimab group. The raw RNA-seq, TCR-seq, and ctDNA-seq reads generated in this paper are deposited in Genome Sequence Archive (GSA) under the accession code HRA001773. The raw sequence data are available under restricted access because of data-privacy laws. Source data are provided with this paper.

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

## Acknowledgements

We would like to thank all the patients and their family members for participating in the study, and thank the investigator in each study site as well as all relevant staff involved in the study. This study was funded by National Nature Science Foundation of China (Approval Number: 82072613); supported by Innovent Biologics, Inc., and cofunded by Eli Lilly and Company.

## Author contributions

J.M.X. and Y.L. contributed to the study design. J.M.X., Y.L., S.J.J., Z.M., and C.M. contributed to paper drafting. Q.X.F., Y.Q.S., L.Y., T.J.C., K.S.G., M.T., X.W.W., C.X.C., N.X., J.X.X., Q.L.G., Y.P.L., T.Z., Y.X.B., W.L., Y.P.Z., G.H.D., D.M., J.D.Z., C.M.B., Y.C.H., W.J.L., L.W., X.C., Y.Y., J.Y.W. and S.J.J contributed to patient enrollment and data collection. Y.Q.W. contributed to statistical analysis. H.Z., Y.W., and Z.M. contributed to data interpretation and medical review. B.P., J.Y.S., and C.M. contributed to biomarker analysis and involved in paper drafting.

## Competing interests

H.Z., Y.W., Z.M., Y.Q.W., B.P., J.Y.S., and C.M. are the staff of Innovent Biologics, Inc. Other authors declare no completing interests.

## Additional information

**Peer review information** *Nature Communications* thanks Bing Su and the other anonymous reviewer(s) for their contribution to the peer review this work. Peer reviewer reports are available.

[1]Department of Gastrointestinal Oncology, The Fifth Medical Center, Chinese PLA General Hospital, Beijing, China. [2]Department of Oncology, The First Affiliated Hospital of Zhengzhou University, Zhengzhou, China. [3]Department of Oncology, The First Affiliated Hospital of Nanjing Medical University, Nanjing, China. [4]Department of Radiotherapy, Nantong Tumor Hospital, Nantong, China. [5]Department of Medical Oncology, Fujian Provincial Hospital, Fuzhou, China. [6]Department of Oncology, The First Affiliated Hospital of Anhui Medical University, Hefei, China. [7]Department of Oncology, The First Affiliated Hospital of Soochow University, Suzhou, China. [8]Department of Chemotherapy, Qilu Hospital of Shandong University, Jinan, China. [9]Department of Medical Oncology, Cancer Hospital, Chinese Academy of Medical Sciences and Peking Union Medical College, Beijing, China. [10]Department of Medical Oncology, The First Affiliated Hospital of Zhejiang University, Hangzhou, China. [11]Department of Medical Oncology, The First Affiliated Hospital of Xi'an Jiaotong University, Xi'an, China. [12]Department of Immunotherapy, Henan Cancer Hospital, Zhengzhou, China. [13]Department of Oncology, The First Hospital of China Medical University, Shenyang, China. [14]Cancer Center, Union Hospital Affiliated to Tongji Medical College of Huazhong University of Science and Technology, Wuhan, China. [15]Department of Gastrointestinal Oncology, Harbin Medical University Cancer Hospital, Harbin, China. [16]Cancer Center, The First Hospital of Jilin University, Changchun, China. [17]Department of Thoracic Oncology, Zhejiang Cancer Hospital, Hangzhou, China. [18]Department of Oncology, Chinese People's Liberation Army General Hospital, Beijing, China. [19]Department of Gastrointestinal Oncology, Guangdong Provincial People's Hospital, Guangzhou, China. [20]Department of Gastroenterology, Liaoning Cancer Hospital, Shenyang, China. [21]Department of Medical Oncology, Peking Union Medical College Hospital, Chinese Academy of Medical Sciences, Beijing, China. [22]Department of Thoracic Surgery I, Yunnan Cancer Hospital, Kunming, China. [23]Department of Oncology, Nanfang Hospital, Southern Medical University, Guangzhou, China. [24]Departmentof Thoracic Medical Oncology, Hunan Cancer Hospital, Changsha, China. [25]Department of Oncology, No. 900 Hospital of The Joint Logistic Support Force, Fuzhou, China. [26]Department of Gastrointestinal Oncology, Gansu Provincial Cancer Hospital, Lanzhou, China. [27]Department of Oncology, Affiliated Hospital of Jining Medical University, Jining, China. [28]Innovent Biologics, Inc., Suzhou, China. [29]These authors contributed equally: Jianming Xu, Yi Li. ✉email: jmxu2003@yahoo.com

