## [Peer Review File · Nature Communications]

Reviewers' Comments:

Reviewer #1:

Remarks to the Author:

The other molecules studied in this space are:

Nivolumab (1st and 2nd)
Pembrolizumab (1st and 2nd)
Tislelizumab (2nd but also 1st)
Carmelizumab (2nd)
Toripalimab (1st line)

This study with Sintilimab is the smallest trial. Also has the lowest response rate and smallest delta in the median survival.

The correlative data are not strong. High wnt with better survival is also curious and does not match the literature. the sentence on line 229 should be reworded and difficult to understand.

similarly, patients with low TMB had better OS is also contradictory to the literature (line 247).

In many instances, there is no correlation where one expected.

Reviewer #2:

Remarks to the Author:

Patients with advanced esophageal cancer have poor prognosis and limited treatment options after the failure of first-line chemotherapy. This randomized, phase II trial aimed to compare the efficacy and safety of the anti-PD-1 antibody (sintilimab) versus chemotherapy in Chinese patients with esophageal squamous cell carcinoma after first-line therapy. Generally, the results of this study are meaningful but not very novel, since three phase III trials comparing anti-PD-1 antibody and chemotherapy in patients with esophageal cancer have been published in the recent two years (KEYNOTE-181, ESCORT, and ATTRACTION-3).

Some specific comments:

1. The Introduction section is not very clear to tell readers the reasons for conducting this trial.
2. Table 1: Regarding baseline characteristics of patients, some important information is missed, such as disease stage after first-line therapy, metastases site, histological grade, and PD-L1 status.
3. Regarding tumor response assessment, what is the definition for measurable lesion? If the primary tumor of esophagus is not measurable, what is the diagnostic criteria for positive lymph node?
4. For NLR, what is the reason for choose the time point of 6 wk after sintilimab treatment?

Reviewer #3:

Remarks to the Author:

Abstract:

Lines 65-67: It would be more appropriate to put the median survival times and have a separate line/comment about the response rates.

Introduction:

Appropriate information and motivation provided.

Results:

Lines 139-140: Are these differences statistically significant?

Line 172: Were there formal comparisons of QoL. Nothing is mentioned in the methods.

Lines 222-223: Is the interaction between these cells and treatment formally assessed? that would strengthen your comment.

Methods:

Design and endpoints are appropriately described.

Line 465: Should specify that the log-rank test is stratified by performance score.

Line 467: Since the primary endpoint (OS) is evaluated at a two-sided alpha of 0.20, would it not be more consistent to also present the 80% confidence interval for the hazard ratio?

Line 470: When comparing the groups at multiple timepoints, some method should be employed to control the family-wise type I error.

Lines 471-472: The comparisons of ORR and DCR should also incorporate the stratification factor - the Cochran Mantel Haenszel test would be more appropriate.

Figures and Supplements:

Supplementary Tables 1 and 2: Are those raw p-values or has some adjustment been made to limit false discovery?

Supplementary Table 4: It should be made clearer that the change from baseline is being presented.

Figures 2 3, and 6 / Supplementary Figure 2: Images came out a little fuzzy.

Figure 3 / Supplement Figure 1: It is not clear what the counts and percentages under the treatment groups represent. The HRs could also include the p-values.

REVIEWER COMMENTS

Reviewer #1 (Remarks to the Author):

The other molecules studied in this space are:

Nivolumab (1st and 2nd)

Pembrolizumab (1st and 2nd)

Tislelizumab (2nd but also 1st)

Carmelizumab (2nd)

Toripalimab (1st line)

This study with Sintilimab is the smallest trial. Also has the lowest response rate and smallest delta in the median survival.

The correlative data are not strong. High wnt with better survival is also curious and does not match the literature. the sentence on line 229 should be reworded and difficult to understand.

In many instances, there is no correlation where one expected.

Authors Reply: We appreciate your comments and indeed, the gene expression correlation analysis did not show results according to previously published literature. Especially the weak prognostic effect of a general T cell infiltration is surprising and might be attributed to the small sample size. Ideally one would further analyze these tumor samples by IHC or mIF to dissect the immune-phenotypes and to assess the location of immune cell infiltration. Unfortunately, no additional tumor samples are available for such subsequent analyses.

Also, the correlation analysis of signaling pathways revealed partially conflicting data: Wnt pathway generally is associated with tumorigenesis, metabolic reprogramming of cancer cells [1] and therefore with worse outcome. High Wnt expression also is associated with poor T cell infiltration and non-T-cell-inflamed tumor microenvironments [2]. However, the function of Wnt pathway might be context-dependent and other findings support a potential role for Wnt/ β -catenin signaling in CD8+ T cell differentiation and memory formation [3,4]. Wnt pathway also enhances the proliferation and cytolytic activity of human $\gamma\delta$ T cells [5]. Autocrine WNT pathway activation induces vascular normalization and endows endothelial cell basement membrane the capacity to boost T cell infiltration [6]. Further evidence shows that, in contract to early stage tumors, advanced and metastatic tumors might have downregulated Wnt expression. In some cases, endogenous Wnt signaling could even counteract tumour progression [7].

We acknowledge the conflicting literature and added additional information and references into the result and discussion section. Also, we revised and expanded the sentence in line 229 of the previous manuscript into lines 233-239 of the revised manuscript. Additionally, we expanded and clarified respective sections of

the discussion in line 337-359.

Reference:

- [1]. El-Sahli S, Xie Y, Wang L, Liu S. Wnt Signaling in Cancer Metabolism and Immunity. *Cancers (Basel)*. 2019 Jun 28;11(7):904.
- [2]. Luke JJ, Bao R, Sweis RF, Spranger S, Gajewski TF. WNT/ β -catenin Pathway Activation Correlates with Immune Exclusion across Human Cancers. *Clin Cancer Res*. 2019 May 15;25(10):3074-3083.
- [3]. Gattinoni L, Zhong XS, Palmer DC, Ji Y, Hinrichs CS, Yu Z, Wrzesinski C, Boni A, Cassard L, Garvin LM, Paulos CM, Muranski P, Restifo NP. Wnt signaling arrests effector T cell differentiation and generates CD8+ memory stem cells. *Nat Med*. 2009 Jul;15(7):808-13.
- [4]. Gattinoni L, Ji Y, Restifo NP. Wnt/beta-catenin signaling in T-cell immunity and cancer immunotherapy. *Clin Cancer Res*. 2010 Oct 1;16(19):4695-701.
- [5]. Chen YQ, Zheng L, Aldarouish M, Zhou ZH, Pan N, Liu JQ, Chen FX, Wang LX. Wnt pathway activator TWS119 enhances the proliferation and cytolytic activity of human $\gamma\delta$ T cells against colon cancer. *Exp Cell Res*. 2018 Jan 1;362(1):63-71.
- [6]. Martínez-Rey D, Carmona-Rodríguez L, Fernández-Aceñero MJ, Mira E, Mañes S. Extracellular Superoxide Dismutase, the Endothelial Basement Membrane, and the WNT Pathway: New Players in Vascular Normalization and Tumor Infiltration by T-Cells. *Front Immunol*. 2020 Oct 30;11:579552.
- [7]. Albuquerque C, Pebre Pereira L. Wnt Signalling-Targeted Therapy in the CMS2 Tumour Subtype: A New Paradigm in CRC Treatment? *Adv Exp Med Biol*. 2018;1110:75-100.

similarly, patients with low TMB had better OS is also contradictory to the literature (line 247).

Authors Reply: Thank you very much for this comment. In this study, we did not test the tumor mutational burden (TMB) as defined as the total number of somatic non-synonymous mutations per coding area of a tumour genome and expressed by mutations/Mb [1-3]. Instead, we ran the correlation analysis between the molecular tumor burden index (mTBI) in relation to clinical outcome. mTBI is a reflection of the percentage of circulating tumor DNA (ctDNA) detected in cell-free DNA (cfDNA) and can reflect the tumor burden at the molecular level. The calculation of mTBI was described first in Wang et al. [4]. We appreciate the risk of confusing TMB and mTBI and clarified this in the Method and Material and the Discussion section. There we describe our findings that mTBI was significantly associated with clinical efficacy in the sintilimab treatment group and, more intriguing, combination of mTBI and TCR clonality could enhance the prediction value.

Reference:

- [1]. Yarchoan M, Albacker LA, Hopkins AC, Montesion M, Murugesan K, Vithayathil TT, Zaidi N, Azad NS, Laheru DA, Frampton GM, Jaffee EM. PD-L1 expression and tumor mutational burden are independent biomarkers in most cancers. *JCI Insight*. 2019 Mar 21;4(6):e126908.
- [2]. Fancello L, Gandini S, Pelicci PG, Mazzarella L. Tumor mutational burden quantification from targeted gene panels: major advancements and challenges. *J Immunother Cancer*. 2019 Jul 15;7(1):183.
- [3]. Marabelle A, Fakih M, Lopez J, Shah M, Shapira-Frommer R, Nakagawa K, Chung HC, Kindler HL, Lopez-Martin JA, Miller WH Jr, Italiano A, Kao S, Piha-Paul SA, Delord JP, McWilliams RR, Fabrizio DA, Aurora-Garg D, Xu L, Jin F, Norwood K, Bang YJ. Association of tumour mutational burden with outcomes in patients with advanced solid tumours treated with pembrolizumab: prospective biomarker analysis of the multicohort, open-label, phase 2 KEYNOTE-158 study. *Lancet Oncol*. 2020 Oct;21(10):1353-1365.
- [4]. Wang Y, Zhao C, Chang L, Jia R, Liu R, Zhang Y, Gao X, Li J, Chen R, Xia X, Bulbul A, Husain H, Guan Y, Yi X, Xu J. Circulating tumor DNA analyses predict progressive disease and indicate trastuzumab-resistant mechanism in advanced gastric cancer. *EBioMedicine*. 2019 May;43:261-269.

Reviewer #2 (Remarks to the Author):

Patients with advanced esophageal cancer have poor prognosis and limited treatment options after the failure of first-line chemotherapy. This randomized, phase II trial aimed to compare the efficacy and safety of the anti-PD-1 antibody (sintilimab) versus chemotherapy in Chinese patients with esophageal squamous cell carcinoma after first-line therapy. Generally, the results of this study are meaningful but not very novel, since three phase III trials comparing anti-PD-1 antibody and chemotherapy in patients with esophageal cancer have been published in the recent two years (KEYNOTE-181, ESCORT, and ATTRACTION-3). Some specific comments:

1. The Introduction section is not very clear to tell readers the reasons for conducting this trial.

Authors Reply: We thank the reviewer for this comment and the introduction section of this manuscript was revised accordingly in line 75-113. The reasons for conducting this trial were to compare the efficacy and safety of sintilimab vs. chemotherapy as second-line treatment for advanced or metastatic ESCC and observe the predictive biomarkers of sintilimab.

2. Table 1: Regarding baseline characteristics of patients, some important information is missed, such as disease stage after first-line therapy, metastases site, histological grade, and PD-L1 status.

Authors Reply: Thank you very much for pointing this out. The baseline

characteristics of patients are updated according to your suggestion. The disease stage of table 1 is referring to clinical TNM stage at time of screening according to AJCC (version 7). The metastases sites and PD-L1 status have been added to table 1. Unfortunately, despite all efforts, the full histological grade of pathological diagnosis could not be collected for all patients since the initial diagnosis was performed mainly in distant local hospitals to which the clinical trial sites don't have access in most cases.

3. Regarding tumor response assessment, what is the definition for measurable lesion? If the primary tumor of esophagus is not measurable, what is the diagnostic criteria for positive lymph node?

Authors Reply: Thank you very much for your question. The definition for measurable lesion is according to RECIST v1.1. Tumor lesions must be accurately measured in at least one dimension (longest diameter in the plane of measurement is to be recorded) with a minimum size of 10mm by CT scan and malignant lymph nodes must be considered pathologically enlarged and measurable, a lymph node must be ≥ 15 mm in short axis when assessed by CT scan.

4. For NLR, what is the reason for choose the time point of 6 wk after sintilimab treatment?

Authors Reply: Thank you very much for your question. The time point for NLR assessment was prospectively set at 6 weeks post cycle 1. This time point was chosen in order to be synchronized with the first tumor assessment. We added this clarification to the manuscript.

Reviewer #3 (Remarks to the Author):

Abstract:

Lines 65-67: It would be more appropriate to put the median survival times and have a separate line/comment about the response rates.

Authors Reply: Thank you very much for your comment. The ORR was deleted in the abstract due to the word limit The manuscript has been revised in the abstract in lines 64-66.

Introduction:

Appropriate information and motivation provided.

Authors Reply: Thank you very much for your comment. The motivation of this study was added in lines 104-113 of the revised manuscript. The Introduction section is revised in lines 75-113 of the revised manuscript.

Results:

Lines 139-140: Are these differences statistically significant?

Authors Reply: Thank you very much for your question. These differences are not significant. RMST is a statistical technique recently adopted widely as an alternative to median to provide an intuitive interpretation of the data. Especially in cases in which the data violate the proportional hazard assumption and when only evaluating median difference could be very misleading, the RMST is more appealing than median survival time. RMST can be viewed as a measure of average survival from time 0 to a specified time point, which reflects a more complete picture of the data over a period of time. Since RMST can be viewed as an alternative to the median estimate, it is more used for descriptive purpose. Here a delayed separation of the survival curve can be observed from the KM plot, therefore we reported RMST. We also provided p-values from the weighted log-rank test. P-values from the Fleming- Harrington (0, 0.2), (0, 0.5) and (0, 1) are less than 0.01, indicating the delayed effect of the significant OS benefit with sintilimab over chemo. We consider the Fleming-Harrington test is more established nonparametric method to do statistical comparison test.

Line 172: Were there formal comparisons of QoL. Nothing is mentioned in the methods.

Authors Reply: Thank you very much for your question. Respective information is now added to the Method and Material section in line 531-534 of revised manuscript.

Lines 222-223: Is the interaction between these cells and treatment formally assessed? that would strengthen your comment.

Authors Reply: Thank you very much for your question. The two signatures of T-follicular helper cells and activated B-cells are moderately correlated (Spearman's correlation=0.48, and $P=2e-08$). This information has been added to the Results section in line 230-232 of revised manuscript. When we build survival model by $Su_{vr} \sim T_{fh} + aB$ for the sintilimab group, the Tfh signal is still significant (HR=0.511, $P=0.025$), but not significant for activated B cells (HR=0.630, $P=0.121$).

Methods:

Design and endpoints are appropriately described.

Line 465: Should specify that the log-rank test is stratified by performance score.

Authors Reply: Thank you very much for your comment. It has been revised that the log-rank test is stratified by performance score.

Line 467: Since the primary endpoint (OS) is evaluated at a two-sided alpha of 0.20, would it not be more consistent to also present the 80% confidence interval for the hazard ratio?

Authors Reply: We acknowledge reviewer's comment regarding to the percentage to be used with CI. It is correct that from statistical perspective, 80%

CI is associated with the original alpha level. However, we think two-sided alpha of 0.05 is more conventionally presented and as descriptive purpose only, just like the cases that CIs reported for interim analysis many times are not adjusted in clinical journals.

Line 470: When comparing the groups at multiple timepoints, some method should be employed to control the family-wise type I error.

Authors Reply: Thank you very much for your comment. RMST was a sensitivity analysis, we don't control type I error for sensitivity analysis.

Lines 471-472: The comparisons of ORR and DCR should also incorporate the stratification factor - the Cochran Mantel Haenszel test would be more appropriate.

Authors Reply: Thank you very much for pointing this out. That's a typo, the CMH was used to compare the relative risk of ORR and DCR. It is corrected in the Methods section in line 527-529 of revised manuscript.

Figures and Supplements:

Supplementary Tables 1 and 2: Are those raw p-values or has some adjustment been made to limit false discovery?

Author response: We appreciate your comment and concerns around multiplicity testing. When adjusting the analysis of multiplicity, no significant correlation could be detected – which also could be due to the small sample number. Since the analysis is meant to be of exploratory nature we decided to use data derived from raw P-values and to put all potential associations out for discussion and further analysis. However, the risk of type I error is acknowledged and for transparency we adjusted the statistical description in the Method section accordingly. Additionally, we rephrased the result section by mentioning the use of raw p values. We thereby hope to adequately balance statistical vigor and discussion of exploratory data.

Supplementary Table 4: It should be made clearer that the change from baseline is being presented.

Author response: Thank you very much for your comment. Table 4 has been updated accordingly.

Figures 2 3, and 6 / Supplementary Figure 2: Images came out a little fuzzy.

Author response: Thank you very much for your comment. Supplementary Figure 3 has been updated accordingly.

Figure 3 / Supplement Figure 1: It is not clear what the counts and percentages under the treatment groups represent. The HRs could also include the p-values.

Author response: We appreciate your comment very much and clarified the figure by adding a description of the numbers and percentages of respective

columns. However, since the analysis of subgroups was performed as sensitivity analysis, we feel it would not be necessary to include the p-values. Also, it seems rather unusual to provide p-values on top of HR and CI as show in the below examples [1,2].

Reference:

- [1]. Kato K, Cho BC, Takahashi M, et al. Nivolumab versus chemotherapy in patients with advanced oesophageal squamous cell carcinoma refractory or intolerant to previous chemotherapy (ATTRACTION-3): a multicentre, randomised, open-label, phase 3 trial. *Lancet Oncol.* 2019 Nov;20(11):1506–1517.
- [2]. Kojima T, Shah MA, Muro K, et al. Randomized Phase III KEYNOTE-181 Study of Pembrolizumab Versus Chemotherapy in Advanced Esophageal Cancer. *J Clin Oncol.* 2020 Dec 10;38(35):4138-4148.

Reviewers' Comments:

Reviewer #1:

Remarks to the Author:

Appreciate the responses, however, the study is seriously limited by the number of specimens/patients and multiple testing and discordant results.

Reviewer #2:

Remarks to the Author:

My comments have been fully addressed.

Reviewer #3:

Remarks to the Author:

Comments were all reasonably addressed.

Reviewer #4:

Remarks to the Author:

In this study, the authors investigated the efficacy and safety of anti-PD-1 antibody (sintilimab) versus chemotherapy in Chinese patients with esophageal cancer in a randomized, phase II trial. It provides quite strong evidence that compared with chemotherapy as second-line treatment, sintilimab significantly improved OS in ESCC patients with less toxicity. It clearly confirms previous studies using sintilimab in ESCC patients. Importantly, the authors found that the TCR clonality plus mTBI in blood and NLR are two new biomarkers of sintilimab response. These studies are convincing, but they could be enhanced even further by addressing the following concerns:

1. Since other studies have compared the effects of anti-PD-1 antibody and chemotherapy on ESCC patients, it is necessary to emphasize the new findings of this study. Generally, PD-L1 is a commonly used biomarker of PD-1 therapy, however, it has no effect in the treatment of ESCC patients. This study provides more relevant biomarkers for the response of PD-1 antibody. These points should be clarified in the manuscript writing to increase the importance of this study.
2. In Fig. 4&6, NLR and Blood TCR/cfDNA are strongly associated with the efficacy of sintilimab. What are the mechanisms of these correlations? Some explanations discussion may be helpful. In addition, recent study of sintilimab in the treatment of lung cancer has shown that MHC-class II is significantly correlated with prolonged PFS, do you see the similar correlation in this case?
3. In line 220-231, the authors stated that they analyzed 28 immune cell populations and identified a significant correlation between Tfh/activated B cells with PFS. Why are Tfh, tertiary lymphoid structures and activated B cells rather than CD8 or CD4 T cells so important to the efficacy of sintilimab in ESCC patients? Some confirmation experiments may be necessary, or at least provide a discussion about these data. In addition, since the results come from a bulk RNA-seq data rather than a single cell sequencing data, it is necessary to clarify the principle of how to identify these immune cells more clearly.

In this study, the authors investigated the efficacy and safety of anti-PD-1 antibody (sintilimab) versus chemotherapy in Chinese patients with esophageal cancer in a randomized, phase II trial. It provides quite strong evidence that compared with chemotherapy as second-line treatment, sintilimab significantly improved OS in ESCC patients with less toxicity. It clearly confirms previous studies using sintilimab in ESCC patients. Importantly, the authors found that the TCR clonality plus mTBI in blood and NLR are two new biomarkers of sintilimab response. These studies are convincing, but they could be enhanced even further by addressing the following concerns:

1. Since other studies have compared the effects of anti-PD-1 antibody and chemotherapy on ESCC patients, it is necessary to emphasize the new findings of this study. Generally, PD-L1 is a commonly used biomarker of PD-1 therapy, however, it has no effect in the treatment of ESCC patients. This study provides more relevant biomarkers for the response of PD-1 antibody. These points should be clarified in the manuscript writing to increase the importance of this study.

Response: Thank you for your in-depth review and valuable comments. The manuscript has been revised to highlight further the need of additional biomarkers in ESCC.

2. In Fig. 4&6, NLR and Blood TCR/cfDNA are strongly associated with the efficacy of sintilimab. What are the mechanisms of these correlations? Some explanations discussion may be helpful. In addition, recent study of sintilimab in the treatment of lung cancer has shown that MHC-class II is significantly correlated with prolonged PFS, do you see the similar correlation in this case?

Response: Thank you for your constructive comments. We further elaborated on

the potential biological mechanisms of the predictive effects of TCR clonality and tumor burden in the discussion and added several new references.

We are currently analyzing the predictive value of MHC-II expression and antigen presenting pathways in multiple datasets obtained from sintilimab clinical trials in several indications. However, this analysis is not concluded yet.

3. In line 220-231, the authors stated that they analyzed 28 immune cell populations and identified a significant correlation between Tfh/activated B cells with PFS. Why are Tfh, tertiary lymphoid structures and activated B cells rather than CD8 or CD4 T cells so important to the efficacy of sintilimab in ESCC patients? Some confirmation experiments may be necessary, or at least provide a discussion about these data. In addition, since the results come from a bulk RNA-seq data rather than a single cell sequencing data, it is necessary to clarify the principle of how to identify these immune cells more clearly.

Response: We appreciate your comments and suggestions. Ideally one would further analyze these tumor samples for deeper characterization of the immune-contexture also and especially with regards to spatial distribution of different immune cell subsets and to identify tertiary lymphoid structures (e.g. via mIF or spatial transcriptomics). Unfortunately, no additional tumor samples are available and we added this to the discussion section.

The 28 immune cell subpopulations were calculated based on a set of pan-cancer metagenes (non-overlapping sets of genes that are representative for specific immune cell subpopulations and are neither expressed in tumor cell nor in normal tissue) as described in reference 28 [Cell Rep. 2017 Jan 3;18(1):248-262].

The emerging importance of activated B cells, follicular T helper cells and tertiary lymphoid structures (even in T cell low tumors) for cancer immunotherapy is supported by recent studies in multiple indications as listed in the discussion section. We added a new reference suggesting an increased pool of B cell clones and tertiary lymphoid structures in ESCC [J Leukoc Biol. 2020 Oct;108(4):1307-1318.]. However, results from this study certainly warrants further investigation